# Response to Bridging Therapy as a Prognostic Indicator of Post-Transplantation Hepatocellular Carcinoma Recurrence and Survival: A Systematic Review

**DOI:** 10.3390/cancers16223862

**Published:** 2024-11-18

**Authors:** Paweł Topolewski, Dariusz Łaski, Martyna Łukasiewicz, Piotr Domagała, Roeland F. de Wilde, Wojciech G. Polak

**Affiliations:** 1Division of Quality in Healthcare, Medical University of Gdańsk, 80-210 Gdańsk, Poland; 2Department of Surgical Oncology, Transplant Surgery and General Surgery, Medical University of Gdańsk, 80-210 Gdańsk, Poland; 3Erasmus MC Transplant Institute, University Medical Center Rotterdam, Department of HPB- & Transplant Surgery, 3015 GD Rotterdam, The Netherlands

**Keywords:** hepatocellular carcinoma, bridging, liver transplantation, transplant oncology, transarterial chemoembolization, thermal ablation, locoregional therapy, Milan criteria

## Abstract

Liver transplantation is one of the most effective treatments for hepatocellular carcinoma, but only when specific transplantation criteria are met. Local therapies may be used to prevent the tumor from exceeding the transplantation criteria (i.e., so-called bridging therapy). However, the impact of bridging therapy on transplantation outcomes and its predictive value are still not known. We performed a systematic review on both radiological and histopathological responses as prognostic indicators of transplantation outcomes. Five studies were included. The overall risk of bias was serious across the studies. If the tumor showed a good radiological response to bridging therapy, there was a greater chance of better transplantation outcomes. Complete tumor necrosis was not associated with better transplantation outcomes. Future predictive models should include radiological, pathological, histological, cellular, and molecular tumor features.

## 1. Introduction

Liver transplantation (LT) is one of the most effective treatments for hepatocellular carcinoma (HCC) in a cirrhotic liver. The prevalence of HCC is increasing worldwide and is the third leading cause of cancer-related death [1,2]. HCC accounts for the vast majority of primary liver cancers [3] and has a median survival of 6–⁠10 months [4]. Patients with HCC are listed for LT according to various transplantation criteria, which may ensure the best overall survival and overall treatment outcome. The most common criteria are the Milan criteria (MC) proposed by Mazzaferro et al. [5]. However, other transplantation criteria, such as the UCSF criteria [6], “up-to-seven” [7], Toronto [8], and Warsaw [9] amongst many other criteria, are associated with excellent post-transplantation outcomes and are accepted for patient selection.

Neoadjuvant treatment for HCC in patients listed for LT is advised but is still debatable due to the low level of evidence [10]. The rationale for neoadjuvant treatment may apply to patients in the setting of (a) bridging where already listed for LT and the goal is to achieve the lowest possible volume of viable tumor in the liver recipient or (b) downstaging with the aim to downsize the tumor load so that the patient can meet the criteria for LT and receive LT. There are various methods for neoadjuvant treatment, which include transarterial chemoembolization (TACE), thermal ablation (i.e., radiofrequency ablation (RFA) or microwave ablation (MWA)), surgical resection, and radiotherapy. Previously, some researchers [11] have aimed to assess the ability of the radiological response evaluation to bridging treatment in order to predict post-transplantation outcomes, however no definitive conclusions were drawn. Hereto, the percentage of viable tumor volume in the explanted liver and radiological response by the Response Evaluation Criteria in Solid Tumors (RECIST) [12] and their modified versions (mRECIST) were analyzed [13].

The aim of this study was to perform a systematic review of the prognostic value of bridging treatment, in terms of radiological and histopathological examination outcomes, on the survival after LT.

## 2. Materials and Methods

### 2.1. Search Strategy

A systematic review was performed according to Preferred Reporting Items for Systematic Review and Meta-Analyses (PRISMA) 2020 guidelines [14]. The research questions are presented in the table listing the Patient, Intervention, Comparison, Outcome (PICO) criteria (Table 1). MEDLINE via the PubMed and Web of Science databases was searched. Additionally, Cochrane Reviews were checked for applicable studies. The keywords used were the following: HCC, hepatocellular carcinoma, liver transplantation, locoregional therapy, transarterial chemoembolization, transarterial radioembolization, selective internal radiotherapy, radiotherapy, bridging, response. The searched terms were combined using the Boolean operators “AND” and “OR”. No filters were used. The full search strategy of MEDLINE via PubMed was the following: ((“Carcinoma, Hepatocellular” [15]) OR “hepatocellular carcinoma” OR “hepatocellular” OR “HCC”) AND (“Liver Transplantation” [15] OR “liver transplantation”) AND ((“local” OR “locoregional”) OR (“TACE” OR “SIRT” OR “TARE” OR “SBRT” OR “Selective Internal Radiation Therapy” OR “chemoembolization” OR “chemoembol*” OR “radioembolization” OR “radioembol*”) OR (“bridging” OR “bridg*” OR “downstaging” OR “downsta*”) OR (“Ablation Techniques” [15] OR “RFA” OR “ablation”) OR (“Radiotherapy” [15] OR “radiotherapy”)) AND (respon*). The full search strategy of Web Of Science was the following: ((“hepatocellular carcinoma” OR “hepatocellular” OR HCC) AND (“liver transplantation” OR “liver transplant”) AND ((“local” OR “locoregional”) OR (“TACE” OR “SIRT” OR “TARE” OR “SBRT” OR “Selective Internal Radiation Therapy” OR “chemoembolization” OR “chemoembol*” OR “radioembolization” OR “radioembol*”) OR (“bridging” OR “bridg*” OR “downstaging” OR “downsta*”) OR (“Ablation Techniques” OR “RFA” OR “ablation”) OR (“Radiotherapy” OR “radiotherapy”)) AND (respon*)). The citations were synthesized using EndNote 21. Duplicates were removed also using EndNote 21. The references of systematic reviews in similar subjects and similar studies were scanned and analyzed for relevant studies that were not found in our search. The study protocol was registered at PROSPERO ID: CRD42024583554.

### 2.2. Study Selection

Inclusion criteria were the following: age > 18 years, deceased donor liver transplantation as final treatment, entire patients’ cohort within standard qualification criteria (Milan, UCSF, up-to-seven) before starting bridging treatment, response to treatment reported by means of mRECIST or RECIST criteria and explant histopathological examination, survival, and recurrence of HCC after LT reported.

Exclusion criteria were the following: study outcomes not reported in English, not a full-text article type (e.g., conference meeting abstract, poster, abstract only, no full text available), cohort including patients undergoing downstaging before starting bridging treatment, no response to treatment reported, no survival after LT reported, or mixed types of final treatment (e.g., surgical resection, thermal ablation, living donor liver transplantation).

We excluded studies that included both bridging and downstaging therapies due to the high risk of interference from mixed neoadjuvant treatment modalities, as was predefined in the PICO questions.

The selection process included reading titles and abstracts, reading full texts, and extracting data by two independent reviewers (PT and MŁ), with disagreements resolved by a third reviewer (DŁ).

### 2.3. Data Extraction

Two independent authors (PT and MŁ) collected relevant data from the literature into spreadsheets for each included study: first author, year published, study timeframe, geographic area of the study, design, center in which the study was conducted, transplantation qualification criteria, study inclusion criteria, number of patients initially undergoing bridging therapy, number of patients who underwent LT, number of tumors included in the study, age, cirrhosis etiology, MELD scores at liver transplantation listing and at transplant, tumor stage before bridging, intervention modalities, control group, time to LT, response to treatment in RECIST or mRECIST criteria, response to treatment assessed in explant pathological outcome, hazard ratio (HR) and 95% confidence interval (CI), recurrent-free survival (RFS) after LT, overall survival (OS) after LT, and follow-up time.

### 2.4. Statistical Methods, Risk of Bias and Literature Quality Evaluation

A meta-analysis of HCC recurrence rates (RRs) was intended or the RFS and OS were stratified by responses based on the percentage of tumor necrosis and RECIST or mRECIST outcomes, which unfortunately was not feasible due to the heterogeneity of the reported outcomes and intervention types. The results of the included studies reporting HRs and other significant differences and outcomes are summarized in Tables 2–4.

The risk of bias was evaluated independently by two authors (PT and MŁ), with disagreements consulted and resolved by a third reviewer (DŁ). ROBINS-I [16] was used to assess risk of bias in the included studies by evaluating the area of bias. The tool was designed for the assessment of bias due to confounding, the selection of the participants, the classification of interventions, the deviations from intended interventions, missing data, the measurement of outcomes, or the selection of the reported results. The risk of bias in these studies was assessed as low-risk, moderate-risk, critical-risk, and no information. The Robvis [17] tool was used to visualize the risk of bias and create summary plots.

## 3. Results

On 18/09/2024, MEDLINE via PubMed and Web of Science was searched. No relevant articles were found in the Cochrane Reviews. A total of 223 duplicates were identified by EndNote and removed. A total of 105 articles was chosen for a full-text screening. Of these, 100 reports were excluded for the following reasons: not compliant with the PICO criteria (*n* = 54), no survival reported (*n* = 20), no radiological response reported (*n* = 18), or no pathological response reported (*n* = 8). Ultimately, five studies [18,19,20,21,22] met the inclusion criteria. The study flowchart was presented in Figure 1.

The included studies were published between 2012 and 2023, and their timeframe ranged from January 1989 to December 2021. All of them were retrospective studies, four were single-center studies, and one was a multicenter study. In all studies, the Milan criteria were used as the LT qualification criterion. However, in a study by Seehofer et al. [18], the UCSF criteria were used from 2000 onwards. Bridging modalities included TACE in all studies and (laparoscopic) microwave ablation (l)MWA, RFA, percutaneous ethanol injection (PEI), irreversible electroporation (IRE), and multiple or mixed types of treatment in one study. The design and key points of the included studies are presented in Table 2.

There were a total of 1197 patients in the included studies. Patient ages ranged from 52 to 67 years. Most patients had HCV-related cirrhosis (*n* = 557), alcoholic liver disease (*n* = 428), HBV-related cirrhosis (*n* = 187), and non-alcoholic steatohepatitis (*n* = 65) as their underlying disease. Ten patients had liver cirrhosis for other reasons and eight patients had cryptogenic cirrhosis. In a study by Seehofer et al., patients with Child–⁠Pugh C cirrhosis were excluded from the analysis. MELD scores ranged from 7 to 21. Bridging interventions included TACE in 961 patients, mixed TACE, and irreversible electroporation in three patients, IRE in two patients, laparoscopic microwave ablation in 25 patients, RFA or percutaneous ethanol injection in 406 patients, and multiple (mixed) modalities in 200 patients. Time on an LT waiting list ranged between <60 days and up to 349 days. The tumor stages from explanted specimens were reported in a study by Seehofer et al.: three patients had T0 tumors, 17 patients had T1 tumors, 14 patients had T2 tumors, and two patients had T3a stage tumors. The full characteristics of the included populations, interventions, and histopathological and radiological response outcomes are presented in Table 3 and Table 4.

Logistic regression analysis was performed in three studies. Seehofer et al. found the RECIST category to be predictive of favorable RFS in patients inside the MC: HR 1.34 (0.54–⁠3.35, *p* = 0.037) in univariate analysis. However, this RECIST category was insignificant in multivariate analysis in this group of patients. In patients outside the MC, the RECIST category was found to be predictive of favorable RFS: HR 3.2 (1.32–⁠7.77) in univariate (*p* = 0.001) and multivariate analyses (0.01). The RECIST category was also tested in the logistic regression model by the researchers both in patients inside the MC, HR 1.01 (0.5–⁠2.07), and outside the MC, HR 1.13 (0.5–⁠2.55), albeit neither of the tests reached statistical significance. Lai et al. performed logistic regression of the means of the mRECIST categories and LT outcomes (defined as post-transplant recurrence or pretransplant tumor-related delisting). On statistical analysis, progressive disease (PD) was found as a significant risk factor for HCC-dependent LT failure, subhazard ratio (SHR) 5.62 (4.10–⁠7.69), and the outcome was statistically significant (*p* < 0.05). The authors also stated that the number of received locoregional therapies (LRTs) was predictive of HCC-dependent LT outcome, 1 LRT SHR 0.51 (0.36–⁠0.74), 2–⁠3 LRTs SHR 0.66 (0.47–⁠0.93), and the outcomes were statistically significant (*p* < 0.05). In a study by Jotz et al., patients with complete tumor necrosis were found to have a lower mortality, HR 2.24 (0.91–⁠5.53), and the results were statistically insignificant (*p* = 0.078).

An evaluation with ROBINS-I resulted in studies classified as the following: moderate risk of bias (*n* = 1) and serious risk of bias (*n* = 4). No study was either classified as having low or critical risk of bias. The risk of bias domain assessment along with summary plots for overall risk of bias assessment across the studies are presented in Figure 2 and Figure 3.

## 4. Discussion

The use of bridging treatment is supported by the literature and international recommendations [10,23]. The quality of evidence supporting bridging as a method of improving LT outcomes and waitlist dropout rates is low [10,23]. The research goal of this study was not to determine the value of bridging as a method of improving LT outcomes but to determine whether the response to bridging could be predictive of LT outcomes. We aimed to assess the response to bridging on the basis of both radiological and histopathological responses. In this study, we were the first to aim to assess both radiological and histopathological responses in combination, as most of the literature had only assessed radiological or histopathological responses. The mRECIST classification for tumor response after locoregional therapy had good performance in predicting tumor necrosis in HCC patients after locoregional therapy [24] and helped reduce interobserver reliability [25]. The outcome of histopathological examination of explanted livers provides important clinical data that can be used for planning further follow-up and potential recurrence treatments. It is also uncertain whether bridging therapy results in improved post-transplant survival and decreased waitlist dropout.

There are various methods of bridging treatment that can be divided into transarterial techniques (TACE, transarterial embolization, transarterial radioembolization, and others), ablation techniques (RFA, MWA, IRE), and stereotactic body radiation therapy. There were no conclusive recommendations from randomized controlled trials on which technique to use. Bridging-to-transplant therapy and strategies are used on the basis of an individual patient’s context and their contradictions. Thermal ablative techniques are used for lesions that are less than three cm in size. However, it was recently shown that locoregional therapies could also be effective for larger tumors [26]. The use of ablative techniques is limited if the tumor location is too close to vital structures (central bile ducts and vessels or adjacent organs). TACE involves the injection of a cytotoxic agent and an ethiodized oil emulsion into the arterial branches that feed the tumor. Transarterial techniques are used for larger tumors, preserved liver function, and the absence of portal vein thrombosis. SBRT is used if patients are not fit for either ablation or transarterial treatment. In our study, TACE was the most common bridging technique, whereas other techniques include ablative techniques (percutaneous/laparoscopic RFA/MWA), PEI, and IRE. In some patients, bridging techniques were combined. The heterogeneity of the treatments performed was high, which was another factor contributing to the high degree of general heterogeneity of the results and possible bias. Patient selection criteria, inclusion criteria, waitlist times, statistical methods, and data reporting also varied between the studies, which was also a significant contributor to heterogeneity. Although many of the selected articles identified the significance of the reported results, the predictive value of the combined RECIST/mRECIST category and histopathological examination for the LT outcome made it difficult to draw conclusions.

The outcomes of this study show that there is still a deficit in knowledge regarding the value of bridging treatment response as a prognostic factor for LT outcome. Seehofer et al. [18] reported that the response to neoadjuvant TACE, rather than the explant necrosis rate, was a prognostic parameter for LT outcome. Beal et al. [19] reported that complete pathologic response had no impact on LT outcome and that patients within the MC did not require serial bridging treatments. The authors suggested that a good response in a radiological examination is adequate to allow close observation with serial imaging prior to LT. Cannon et al. [20] focused on the technique of IRE and its application as an LT bridging method. The authors reported both a high percentage of tumor necrosis in explanted livers and no recurrences, with 80% three-year OS. Lai et al. [21] performed an extended analysis of a large multicenter database from the European Hepatocellular Cancer and Liver Transplantation Study Group and used the inverse probability of treatment weighting to artificially increase the sample size. The authors concluded that patients with a poor response to bridging treatment were at greater risk for pretransplant tumor-related delisting and post-transplant tumor recurrence. Additionally, the authors stated that bridging treatment was valuable when considering an intention-to-treat approach and that the beneficial effect disappeared when bridging was performed more than four times. Jotz et al. [22] investigated the effect of TACE as a bridge to LT. The authors concluded that total tumor necrosis was associated with improved survival at five years after LT. The outcomes of the presented research are not unambiguous. From the presented results, we can conclude that a good response (defined as an mRECIST category of complete or partial response) is a good predictor of favorable LT outcome in terms of OS and HCC recurrence. The data regarding complete pathological response and prognostic value were contradictory. Grąt et al. [27] performed a large analysis of patients who underwent neoadjuvant treatment (both bridging and downstaging) before LT and reported that complete pathologic response was not associated with favorable long-term LT outcomes. This finding was confirmed by Kang et al. [28], who reported that patients who received neoadjuvant treatment were at significantly greater risk of recurrence than treatment-naïve patients were, despite achieving a complete pathological response. Biologically, Xu et al. [29] described the association of neoadjuvant therapy-induced necrosis with increased lymphangiogenesis and increased risk of post-transplantation lymphatic metastases. In conclusion, a good response to bridging treatment can be a positive prognostic factor of LT outcome. However, a complete pathological response cannot be a positive prognostic factor, and thus, repeating bridging locoregional therapies with the intention of achieving a complete pathological response is questionable.

This study had some notable limitations. The overall risk of bias was assessed as serious in most of the studies, which underlines the need for the careful interpretation of the presented results. Additionally, due to the heterogeneity of the results, the methods of bridging treatment and data reporting did not allow a meta-analysis to be performed and hence contributed to a decreased level of presented evidence. All the included studies were of retrospective designs, only one was a multicenter study, and all studies had rather wide timeframes. The reported results of histopathological examination also varied across the studies, contributing to greater heterogeneity in the results and the inability to perform a meta-analysis. None of the studies focused primarily on the prognostic value of the response to bridging treatment as a prognostic indicator of LT outcome. Our analysis focused on the overall effect of LT bridging. To the best of our knowledge, only one systematic review has been published on a similar subject [11]. The authors of this specific analysis analyzed only radiological outcomes and used a similar methodology but were unable to perform a meta-analysis.

The outcomes of this systematic review provide additional evidence in favor of the careful use of bridging treatment to LT. Kostakis et al. [30] performed a systematic review on bridging therapies, focusing on their survival and waitlist dropout benefits. The outcomes of their work established the importance of bridging in LTs, with the main argument for bridging considering an unknown waitlist time for LT. The results of our analysis also suggest that favorable LT outcomes are greatest when there is a radiological response to treatment, which is defined as a complete response or partial response. A poor response to treatment was generally associated with poor LT outcomes. There are not enough data to support the approach to achieving a complete pathologic response, as it was not proven in most of the studies to be associated with significantly better LT outcomes. Bridging therapies for HCC seem to become a method of choice in patients who are waitlisted, as the number of patients qualifying for LT grows, also considering that for more indications, LT has become a holy grail. The current prognostic value of both radiological and pathological response is limited. However, it can still provide important information for clinical decision making. Future perspectives for predicting LT should include biochemical markers and tumor biology, as understanding HCC biology and molecular genetics holds significant promise for advancing more personalized approaches to patients and improving pre- and post-LT oncological treatment outcomes [31]. Pre-LT oncological treatment contributes to cellular tumor changes and impacts LT outcomes. Atanasov et al. [32] evaluated the presence of tumor necrosis and the frequency levels of angiopoietin and monocyte/macrophage subtypes in recipients’ livers prior to LT and their associations with LT outcome. Both the presence of infiltrating monocyte/macrophage subsets and the related angiopoietin axis were associated with worse OS and RFS after LT, and patients who received TACE as LT bridging had a significantly increased presence of monocytes/macrophages and reduced post-LT HCC recurrence.

Predicting locoregional treatment response is also an important topic in liver oncology. Novel models like the AFP-DIAM [33], AFP-R [34], R3-AFP score [35], and French AFP score [36] are being developed to ensure the most precise therapy selection for each transplant candidate. The uses of biopsy and analysis of the histological features of a tumor also bring valuable information that can be used for clinical decision making. Currently, the role of the HCC microenvironment is being discussed and brings important clinical implications [37,38]. The study of the immunological microenvironment is crucial, as the role of systemic therapies in liver transplantation bridging and downstaging is being discussed more frequently [39,40]. The alpha-fetoprotein (AFP) concentration is also a marker of the response to locoregional treatment and LT. Masior et al. [41] created a predictive model for achieving total tumor necrosis (complete pathologic response) on the basis of the initial AFP concentration and its dynamic after the first TACE session. Achieving total tumor necrosis was not a significant prognostic factor for favorable LT outcomes in our analysis but this model can still be used as an indicator of a good TACE response. Other studies also indicated that the analysis of AFP dynamics before LT and during bridging treatment should play an important role in future predictive models [42,43]. These data show the importance of including radiological, pathological, histological, cellular, and molecular tumor features in future predictive models aimed at predicting LT outcomes. Understanding HCC biology and expanding the possibilities of bridging therapies and targeting oncogenic pathways are warranted to provide an optimal bridging strategy for the best LT outcomes [44].

## 5. Conclusions

The results of our analysis indicated that favorable LT outcomes were greatest when there was a radiological response to treatment, defined as a complete response or partial response. Poor radiological response or progressive disease during bridging treatment was generally associated with worse outcomes after LT. There are not enough data to support the approach to achieving a complete pathologic response, as it was not proven in most of the studies to be associated with significantly better LT overall survival. Radiological, pathological, histological, cellular, and molecular tumor features should be included in future predictive models aimed at LT qualification models.

## Figures and Tables

**Figure 1 cancers-16-03862-f001:**
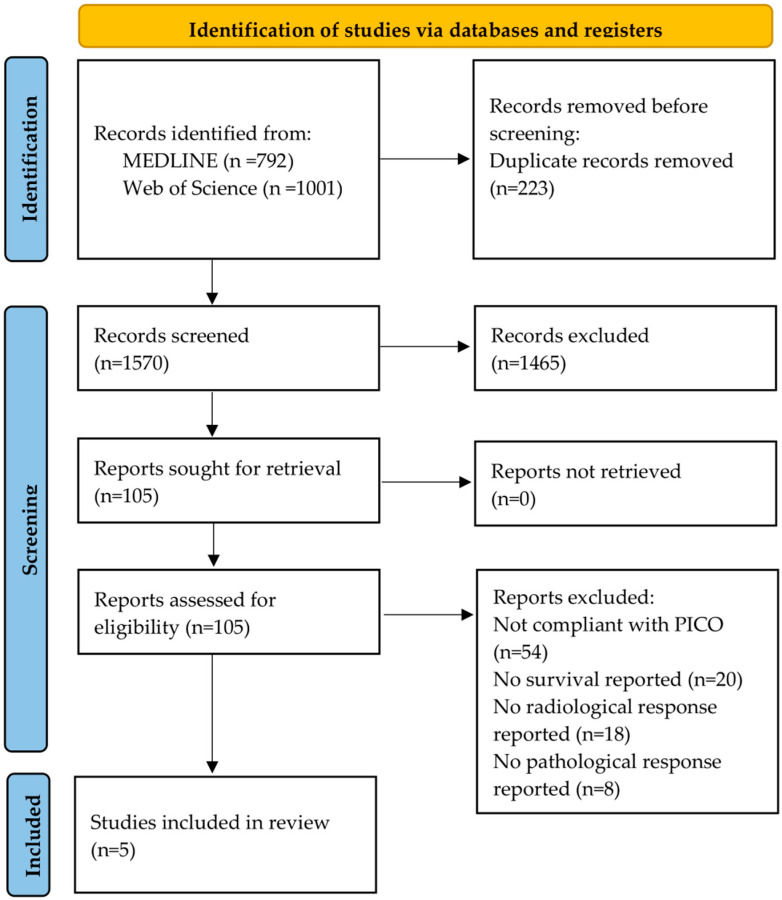
PRISMA flow chart of the literature search and article selection.

**Figure 2 cancers-16-03862-f002:**
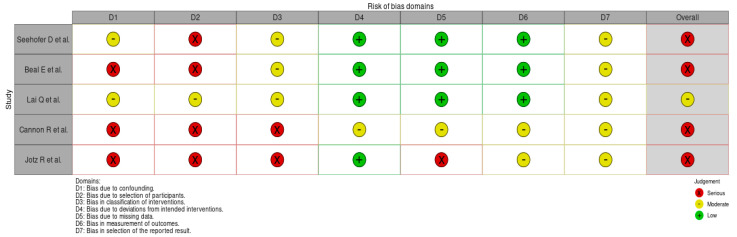
Risk of bias assessment in the included studies. ROBINS-I tools were used, and studies were classified as having low, moderate, serious, or critical risk of bias in seven domains. One study was classified to be at moderate risk of bias and four studies were classified at serious risk of bias. The overall bias was assessed as moderate if the study was judged to be at low or moderate risk of bias for all domains. The overall bias was assessed as serious risk of bias if the study was judged to be at serious risk of bias in at least one domain, but not at critical risk of bias in any domain.

**Figure 3 cancers-16-03862-f003:**
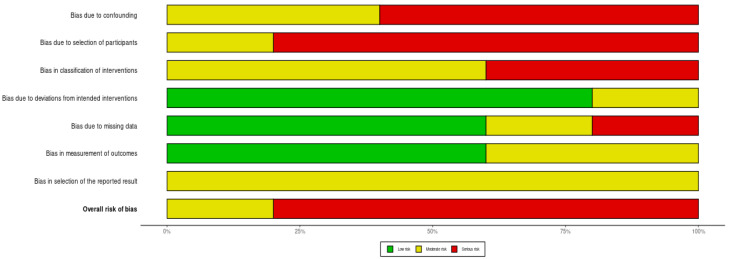
Summary plot of the risk of bias domains for the included studies. ROBINS-I tools were used, and studies were classified as having low, moderate, serious, or critical risk of bias in seven domains. The domain classified at low risk of bias is comparable to a well-performed randomized trial with regard to this domain. The domain classified at moderate risk of bias is sound for a non-randomized study with regard to this domain but cannot be considered comparable to a well-performed randomized trial. The main domains of bias contributing to serious risk of bias were biases due to confounding, selection of participants, and classification of interventions.

**Table 1 cancers-16-03862-t001:** PICO criteria used in this study.

Patient	Patients listed for deceased donor liver transplantation within standard qualification criteria.
Intervention	Bridging locoregional therapy regardless of its type.
Comparison	No bridging of locoregional therapy or no comparison.
Outcome	Response to treatment in RECIST ^1^ or Mrecist ^2^ criteria and explant histopathological examination, recurrence free-survival, and overall survival.

^1^ response evaluation criteria in solid tumors. ^2^ modified response evaluation criteria in solid tumors.

**Table 2 cancers-16-03862-t002:** Summary of the design and key points of the included studies.

First Author and Year Published	Study Timeframe and Geographic Area	Design	Qualification Criteria	Inclusion Criteria	Intervention and Control Group	Key Conclusions
Seehofer D 2012 [18]	January 1989–⁠December 2008 Germany	Retrospective, single-center	Milan (to 2000), UCSF ^1^ (from 2000)	Transplanted, within Milan/UCSF	TACE ^2^, no bridging treatment	The good response to TACE in RECIST ^3^ criteria was a good indicator of recurrent-free survival.
Beal E 2016 [19]	January 2008–⁠31 July 2015 USA	Retrospective, single-center	Milan	Transplanted, within Milan	TACE, lMWA ^4^, no bridging treatment	The presence or absence of viable tumor was not associated with overall survival.
Lai Q 2019 [20]	January 2001–⁠December 2015 Europe	Retrospective, multicenter	Milan	Transplanted, within Milan	TACE, RFA ^5^, PEI ^6^, multiple types, no bridging treatment	A poor radiological response to bridging treatment was associated with a higher risk of recurrence.
Cannon R 2019 [21]	April 2015–⁠June 2016 USA	Retrospective, single-center	Milan	Transplanted, within Milan, IRE as a bridging modality	IRE, TACE combined with IRE ^7^, NA ^8^	No evidence of recurrence in all patients. IRE showed promise as a bridge to liver transplantation for high-risk HCC ^9^.
Jotz R 2023 [22]	January 2013–⁠December 2021 Brazil	Retrospective, single-center	Milan	Transplanted, within Milan, TACE as a bridging modality	TACE, NA	Complete tumor necrosis appeared to be associated with improved patient survival.

^1^ university of california san francisco; ^2^ transarterial chemoembolization; ^3^ Response Evaluation Criteria in Solid Tumor; ^4^ laparoscopic microwave ablation; ^5^ radiofrequency ablation; ^6^ percutaneous ethanol injection; ^7^ irreversible electroporation; ^8^ not applicable; ^9^ hepatocellular carcinoma.

**Table 3 cancers-16-03862-t003:** Summary of group characteristics, intervention modalities, and liver transplantation wait times.

First Author	No. of Patients Initially Undergoing Bridging/Liver Transplantation	No. Tumors	Age	MELD ^1^ Score	No. of Patients Undergoing Each Intervention	Time to Liver Transplantation
Seehofer D [18]	71/71	Single tumor (41) 2–⁠3 tumors (19) 4–⁠5 tumors (4) > 5 tumors (7)	NR ^2^	NR	TACE ^3^	187 days (std. 28) (mean)
Beal E [19]	43/43	1 (median)	56.42 (mean)	13.83 (at transplant) (mean)	TACE (18), lMWA ^4^ (25)	242 days (mean)
Lai Q [20]	942 ^*^/942 ^*^	1 (IQR ^5^ 1–⁠2) (median)	58 (52–⁠63) (median)	12 (9–⁠15) (at diagnosis) (median)	TACE (736), RFA ^6^ or PEI ^7^ (406), multiple (200)	349 days (std. 37.0) (in 2001–⁠2009) (mean)
Cannon R [21]	5/5	8	63 (55–⁠67) (median)	13 (7–⁠21) (at listing) (median)	IRE ^8^ (2), TACE, and IRE (3)	142 days (47–⁠264) (median)
Jotz R [22]	136/136	NR	61.5 (std. 7.0) (mean)	NR	TACE	<60 days

^*^ post-inverse probability of treatment weighting (IPTW) population; ^1^ model for end-stage liver disease; ^2^ not reported; ^3^ transarterial chemoembolization; ^4^ laparoscopic microwave ablation; ^5^ interquartile range; ^6^ radiofrequency ablation; ^7^ percutaneous ethanol injection; ^8^ irreversible electroporation.

**Table 4 cancers-16-03862-t004:** Summary of radiological and histopathological responses to the bridging treatment, recurrence, and survival outcomes in the included studies.

First Author	RECIST ^1^ or mRECIST ^2^ Outcome	Explant Pathological Outcome	Tumor Grade	Microvascular Invasion Presence	Recurrence Outcome	OS ^3^ Outcome
Seehofer D [18]	RECIST: CR ^4^ and PR ^5^ (18) SD ^6^ (35) PD ^7^ (18)	CN ^8^ (13) > 90% necrosis (9), partial necrosis (30), < 10% necrosis (19)	G1 (7) G2 (38) G3 (13) Not assessed (13)	16	Bridging group recurrence rate: 23% No bridging recurrence rate: 29% (*p* > 0.05) Milan-IN: PD vs. CR+PR+SD *p* = 0.352 Milan-OUT: PD vs. CR+PR+SD *p* = 0.047	5-year OS in bridging group 73%. 5-year OS in non-bridging therapy group 67%. Higher OS in bridging group (*p* = 0.522). Higher OS in bridging group in Milan-IN patients (*p* = 0.99). Lower OS in bridging group in Milan-OUT patients (*p* = 0.831).
Beal E [19]	mRECIST: CR in TACE ^9^ group (9), CR in lMWA ^10^ group (20)	CN in TACE ^9^ group (12), CN in lMWA^10^ group (20)	G1 (3) G2 (19) G3 (7)	NR	One recurrence reported	OS was equivalent in both groups (*p* = 0.575). The CN was not associated with OS.
Lai Q [20]	mRECIST: CR (253), PR (263), SD (147), PD (275)	CN (81)	NR ^11^	134	Recurrence in 79 patients	Not reported for the bridging therapy group separately.
Cannon R [21]	RECIST: CR (2), PR (2) NR in 1 patient	CN (2), > 90% necrosis (2), 50% necrosis (1)	NR	0	No recurrences reported	OS 80%.
Jotz R [22]	mRECIST: CR (70), PR (30), SD/PD (10), not reported in 8 patients	CN (76), partial necrosis (29), no necrosis (13)	NR	14	Four recurrences reported	1-year OS 87.3%. 2-years OS 82.1%. 3-years OS 80.9%. 5-years OS 77.5%.

^1^ response evaluation criteria in solid tumors; ^2^ modified response evaluation criteria in solid tumors; ^3^ overall survival; ^4^ complete response; ^5^ partial response; ^6^ stable disease; ^7^ progressive disease; ^8^ complete necrosis; ^9^ transarterial chemoembolization; ^10^ laparoscopic microwave ablation; ^11^ not reported.

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
