# Peer review of "Response to Bridging Therapy as a Prognostic Indicator of Post-Transplantation Hepatocellular Carcinoma Recurrence and Survival: A Systematic Review"

_cancers, 2024, doi:10.3390/cancers16223862_

Round 1
Reviewer 1 Report
Comments and Suggestions for Authors
There are some comments.
It would be better to add an analysis for underlying liver diseases and the presence of liver cirrhosis.
It would be better to add histological features (e.g. histological grade, subtype).
It would be better to confirm the accuracy of the tables.
For example 56,42 -> 56.42 (average age).
It would be better to add detailed legends for Figures 2 and 3.
Comments on the Quality of English LanguagePlease check English grammar and spelling.
Author Response
Comment 1: It would be better to add an analysis for underlying liver diseases and the presence of liver cirrhosis.
Response 1: Thank you for a valuable comment.
We included analysis of underlying liver disease and the presence of liver cirrhosis. [Page 6, lines 177-181]
Comment 2: It would be better to add histological features (e.g. histological grade, subtype).
Response 2: Thank you for a valuable suggestion.
We added details of histological features. Unfortunately, histological HCC subtypes were not available in the articles. We included tumor grading and microvascular invasion presence in the Table 4. [Page 5 and 6, Table 4]
Comment 3: It would be better to confirm the accuracy of the tables. For example 56,42 -> 56.42 (average age).
Response 3: Thank you for pointing that out.
We modified the presentation of the results in Tables 3 and 4 according to your suggestions. [Tables 3 and 4 on pages 6 and 7]
Comment 4: It would be better to add detailed legends for Figures 2 and 3.
Response 4: Thank you for your valuable suggestion.
We provided detailed legends for Figures 2 and 3 to enhance the presentation and understanding of the results. [Figures 2 and 3 on page 8.]
Comment 5: Please check English grammar and spelling.
Response 5: Thank you for pointing that out.
The grammar and spelling in the manuscript was revised by an English native speaker.
Reviewer 2 Report
Comments and Suggestions for Authors
The paper titled "Response to bridging therapy as a prognostic indicator of post-transplantation hepatocellular carcinoma recurrence and survival: a systematic review" provides an adequate systematic review on the impact of bridging therapies on liver transplantation (LT) outcomes for hepatocellular carcinoma (HCC).
While the use of bridging therapy to improve LT outcomes is well-documented, the paper attempts to focus specifically on whether the radiological and histopathological response to such therapies can predict post-transplantation outcomes. The novelty lies in combining both radiological and histopathological outcomes for the first time to evaluate their prognostic value comprehensively. This integrated approach could offer a fresh perspective on optimizing treatment strategies for HCC patients awaiting LT, a significant aspect in liver oncology.
The true strength of the study is the optimal methodology. The authors strictly follows the PRISMA guidelines, ensuring a systematic and transparent approach to the literature search.
However, the paper's conclusions reinforce existing knowledge about the benefits of bridging therapy, particularly when there is a favorable radiological response. Its novelty is somewhat diminished due to the limited new insights: the paper seems to reiterate known facts rather than pushing the boundaries of the field, limiting its contribution in terms of advancing clinical practice.
Indeed, a notable limitation is the inability to conduct a meta-analysis due to the heterogeneity of the studies included. This indicates that the data were too varied in terms of study design, intervention methods, and outcomes, which hampers the ability to draw stronger statistical inferences. The ROBINS-I tool was used to assess bias, with most studies showing a serious risk of bias, further weakening the statistical power of the review. The lack of randomized controlled trials also limits the ability to provide higher-level evidence on the subject. Moreover, an important limitation is the absence of consideration for more comprehensive transplant criteria such as the AFP French score, which combines tumor size, number, and alpha-fetoprotein levels to better stratify patients' transplant eligibility and outcomes.
Author Response
Thank you for a thorough review of our work.
As requested we added an annotation of AFP French score and other novel models of HCC recurrence in the discussion section. [Page 11, lines 365-368]
Reviewer 3 Report
Comments and Suggestions for Authors
The authors present a nice paper abouyt bridging therapy as a prognostic indicator of post- 2 transplantation hepatocellular carcinoma recurrence and survival
Bridging therapy is a technique commonly used by liver team to halve tumor growth and allow patient to fullfilg liver transplamnt criteria.
The research is well conducted and M&M are solid according to the prisma guidelines and PICO criteria.
The last was, unfortunately the responsible for a low N (just 5 articles), however the study is solid in its approach, with good statistics, supported by good figures.
Regarding the pathology description of the specimen I think that the TNM staging would be valuable - is this information availabe in the articles
One picture documenting a good RECIST/mRECIST response would add quality to the manuscript.
In the discussion, a small paragraph reporting the possibile use of biopsies and a complement for the study of HCC and the possibility of predicting TACE response - there are some papers about this
references are updated and ok
Author Response
Thank you for your valuable review of our manuscript.
Comment 1: Regarding the pathology description of the specimen I think that the TNM staging would be valuable - is this information available in the articles
Response 1: Thank you for pointing that out.
We agree with your comment; however, the T stage was reported in only one study, and we have included this information in the results section. In the other studies, TNM staging was not reported. [Page 6, lines 185-188.]
Comment 2: One picture documenting a good RECIST/mRECIST response would add quality to the manuscript.
Response 2: Thank you for your valuable comment.
We agree with your suggestion; however, as this is a systematic review, we do not present an image illustrating the response according to RECIST criteria after treatment. The included studies contain only figures showing the response according to RECIST criteria. Therefore, we are unable to provide the suggested figure.
Comment 3: In the discussion, a small paragraph reporting the possibile use of biopsies and a complement for the study of HCC and the possibility of predicting TACE response - there are some papers about this
Response 3: Thank you for the valuable suggestion.
We I have therefore included a brief discussion on the use of biopsy, tumor microenvironment study, and TACE response prediction in the context of liver transplantation bridging. [Page 11, lines 365-373.]
Round 2
Reviewer 1 Report
Comments and Suggestions for Authors
The manuscript was well-revised.
There is a minor comment.
It would be better to adjust the line spacing in Table 4 to make it easier to distinguish.
Comments on the Quality of English LanguageIt would be better to check Enlgish grammar and spelling.
e.g., Micro-vascular Invasion presence -> Presence of Micro-vascular Invasion